# Exposing Cross-Platform Coordinated Inauthentic Activity in the Run-Up to the 2024 U.S. Election

## ABSTRACT

Coordinated information operations remain a persistent challenge on social media, despite platform efforts to curb them. While previous research has primarily focused on identifying these operations within individual platforms, this study shows that coordination frequently transcends platform boundaries. Leveraging newly collected data of online conversations related to the 2024 U.S. Election across 𝕏 (formerly Twitter), Facebook, and Telegram, we construct similarity networks to detect coordinated communities exhibiting suspiciously similar sharing behaviors within and across platforms. Introducing an advanced coordination detection model, we reveal evidence of potential foreign interference, with Russian-affiliated media being systematically promoted across Telegram and 𝕏. Our analysis also uncovers substantial intra- and cross-platform coordinated inauthentic activity, driving the spread of highly partisan, low-credibility, and conspiratorial content. These findings highlight the urgent need for regulatory measures that extend beyond individual platforms to effectively address the growing challenge of cross-platform coordinated influence campaigns.

**ACM Reference Format:**
Anonymous Author(s). 2025. Exposing Cross-Platform Coordinated Inauthentic Activity in the Run-Up to the 2024 U.S. Election. In *Proceedings of Proceedings of the ACM Web Conference 2025 (WWW '25)*. ACM, New York, NY, USA, 18 pages. https://doi.org/10.1145/nnnnnnn.nnnnnnn

## 1 INTRODUCTION

Social media has evolved into a vital arena for public discourse, serving as a platform where individuals and communities can converge to discuss political, social, and cultural issues. These platforms have been instrumental in fostering large-scale movements, such as the Arab Spring and Black Lives Matter, where activists and citizens alike have used social media to amplify calls for justice and change [17, 21]. However, social media has also been implicated in driving polarization and radicalization, as exemplified by the events surrounding the U.S. Capitol attack [12, 31], where online platforms played a notable role in mobilizing and coordinating individuals with extremist views [57]. Similar cases include the Christchurch mosque shootings [14] and far-right [26, 52] rallies across Europe, where social media content has been shown to exacerbate divisive sentiments and mobilize fringe communities [10]. The influence of

online discourse has led to the emergence of state-sponsored information operations aimed at steering public opinion by introducing false or misleading narratives on popular platforms [40, 43, 60].

Coordinated and inauthentic behavior represents a prominent tactic for distorting public discourse by amplifying specific viewpoints and giving the illusion of widespread support [8, 19, 25, 41, 50]. These activities typically involve actions such as synchronized posting and co-activity behavior patterns [35, 47, 50]- such as similar retweets, hashtag sequences, link sharing- designed to push particular narratives to the forefront of public attention. Studies have also documented how these coordinated behaviors contribute to a range of detrimental social effects, such as the spread of propaganda [23], conspiracy theories [55], and the promotion of disinformation [59]. Moreover, coordinated online actions have been linked to heightened toxicity in online conversations [28] and the dissemination of extremist ideologies [27, 57].

In recent years, a growing body of research has focused on the role of social media in the context of political elections. For example, studies on the Brazilian elections have documented the strategic use of bots and coordinated networks to shape public opinion and influence voter perceptions, highlighting the risks of computational propaganda in electoral processes [48]. Similarly, research on European elections has illustrated how misinformation campaigns spread across social media can alter public perceptions, as seen in countries such as France, Germany, and Italy [6, 15, 29, 45]. In the United States, coordinated online behaviors around elections have garnered significant attention, with investigations revealing how coordinated bot activities and foreign information operations have sought to manipulate voter beliefs and amplify divisive content across multiple election cycles [7, 22, 33, 55].

### Contribution of this work

In this work, we expand on current research in coordination detection by broadening our scope to include the identification of coordinated inauthentic activity (CoIA) spanning multiple platforms. We build on, adapt, and advance state-of-the-art coordination detection techniques to identify intra- and cross-platform CoIA that promotes external web domains and amplify specific narratives in the context of the 2024 U.S. Election. We leverage a large-scale dataset covering election-related online conversations spanning several platforms, including Twitter/𝕏, Facebook, and Telegram, in the run-up to the Presidential Election. Combining insights from computational techniques for coIA detection and AI models for content analysis, we aim to answer the following Research Questions (RQs):

**RQ₁:** **Web Domain Promotion.** *Do we observe intra- and cross-platform CoIAs aimed at redirecting traffic towards specific web domains? What are the characteristics of coordinated accounts and which specific domains they promote?*

**RQ$_2$: Content Amplification.** *Do we observe CoIAs pushing specific narratives? What are the characteristics of coordinated accounts and which specific topics they amplify?*

**RQ$_3$: Engagement & Impact.** *What level of engagement do CoIAs generate across various platforms? And how does it compare to the engagement garnered by organic users?*

Leveraging our multi-platform dataset and advancing coIA detection techniques, we uncover multiple networks of coordinated inauthentic accounts. We analyzed the textual content and link-sharing behaviors of these accounts, identifying the narratives they aim to amplify and the external domains they direct traffic to, assessing both their credibility and ties to foreign and domestic entities. Our findings reveal coordinated efforts to promote Russian-affiliated media across Telegram and $\mathbb{X}$, with highly partisan, low-credibility content systematically amplified by these networks. Conspiracy theories surrounding public health, the environment, and political topics such as immigration and geopolitical tensions were prominently featured. Notably, QAnon-related narratives were especially prevalent on Telegram, with coordinated accounts driving much of the discussion. Furthermore, we assessed the prevalence of AI-generated content produced by coordinated actors and the level of engagement their content attracted. We found that coordinated actors on Telegram relied on AI-generated content significantly more than organic users, while the opposite trend was observed on Facebook. This study sheds new light on cross-platform coordination efforts related to the upcoming U.S. election, revealing the complex dynamics of influence operations. These findings underscore the urgent need for regulatory measures that go beyond individual platforms to effectively tackle the growing challenge of cross-platform coordinated inauthentic activity.

## 2 RELATED WORK

### 2.1 Influence campaigns interfering with US Elections

A growing body of research has focused on the influence of coordinated disinformation campaigns during U.S. elections, particularly through the manipulation of social media platforms [16, 54]. Studies have uncovered significant efforts by foreign and domestic actors to manipulate the political landscape during both the 2016 and 2020 U.S. Presidential Elections [7, 57]. For example, the Russian Internet Research Agency (IRA) was linked to a large-scale disinformation campaign during the 2016 election, deploying thousands of social bots and state-sponsored trolls to promote divisive narratives and foster discord [4, 5]. The activity of these inauthentic actors has been extensively studied on Twitter [2, 32, 34, 62], with fewer studies examining their presence on other platforms [63, 64].

Similarly, the 2020 election witnessed the propagation of various false claims and conspiracy theories, including narratives about voter fraud and COVID-19 misinformation [16]. Prominent examples include the "Stop the Steal" movement, which spread across platforms like Twitter and Facebook, inciting allegations that the election results were fraudulent [12]. Coordinated activity surrounding this narrative was widespread, often involving amplification techniques such as automated retweets or sharing similar content across accounts to create an illusion of widespread consensus [31].

### 2.2 Detection of Coordinated Inauthentic Activity

Coordinated inauthentic activity has been identified as a predominant tactic in spreading disinformation and conspiratorial content [35, 49]. Researchers have developed sophisticated methods to detect coordination. Coordination is not limited to malicious campaigns; it also encompasses various social movements and legitimate organizing efforts. However, in the context of disinformation campaigns, coordinated behavior is typically characterized by manipulative actions aimed at amplifying false narratives [44, 50].

The detection of such deceptive, orchestrated efforts has evolved to address both automated and human-coordinated activities through advanced machine learning and network-based approaches. Machine learning techniques have traditionally focused on identifying bot characteristics, such as posting frequency and network behavior, and distinguishing them from human behavior [9, 61]. Recently, attention has shifted toward detecting human-operated accounts, with research emphasizing content-, behavioral-, and sequence-based methods to detect coordinated actions by state-sponsored trolls [3, 13, 24, 30, 34, 46].

Network-based detection, however, has gained prominence due to its ability to reveal coordinated behavior by constructing networks that highlight similarities in user actions, such as shared content, hashtags, or synchronized posting times [35, 37, 38, 44, 49, 50]. These networks are analyzed using properties like node centrality and edge weight, which help identify clusters of users involved in coordinated IOs [35, 56, 57]. This approach has proven effective in uncovering both automated and human-coordinated activities, providing critical insights into the tactics and structure of influence campaigns.

## 3 DATA

**Collection.** The rationale behind our data collection is to capture the online discourse surrounding the 2024 U.S. Presidential Election across multiple social media networks. The data collection for our study spans May and June 2024 and covers three major online platforms: Facebook, $\mathbb{X}$ (formerly Twitter), and Telegram. Each platform was queried to obtain posts or messages containing specific election-related keywords. The full list of keywords used to filter and collect data is provided in Appendix A.

Facebook data was collected through (the now defunct) Crowdtangle, which offered access to public posts of groups and pages. For $\mathbb{X}$, we gather publicly available information, including original tweets, retweets, replies, and quotes, retrieved via the platform's web interface. Finally, Telegram data was gathered using the Telegram API, allowing the extraction of public chats' details, meta data, messages, and message attachments. Details on the $\mathbb{X}$ and Telegram data collection infrastructure can be found in [1].

**Statistics.** For each platform, the collected content consists of posts on Facebook, tweets on $\mathbb{X}$, and messages on Telegram. To establish potential coordination among accounts operating on distinct platforms, we examine the co-sharing of URLs, as they represent specific pieces of information that can be easily tracked across different platforms [36]. In fact, URLs can be embedded in user's posts on each platform under analysis. These URLs serve as a common

thread linking sharing activities across the Web. Thereby, we identify and count unique URLs embedded within the shared content of each dataset. Table 1 presents a summary of the datasets, including the number of accounts,[1] posts, and unique URLs for each platform.

| Platform | Pages | Posts | URLs | Domains |
|---|---|---|---|---|
| Facebook | 6,137 | 46,310 | 15,009 | 5,247 |
| Platform | Accounts | Tweets | URLs | Domains |
| X/Twitter | 178,379 | 6,021,428 | 582,052 | 35,922 |
| Platform | Channels | Messages | URLs | Domains |
| Telegram | 15,537 | 4,309,880 | 2,087,078 | 183,924 |

**Table 1: Datasets statistics.**

## 4 METHODOLOGY

### 4.1 Detecting CoIA across social media platforms

Coordinated accounts can execute their campaigns using various strategies; here, we analyze campaigns that focus on promoting specific URLs (or web domains) and campaigns that produce highly similar textual content to amplify certain topics. The former strategy is commonly used to create the illusion of public consensus around certain viewpoints by artificially amplifying links to external webpages, mock websites, and other social media networks [18, 39]. The latter is often employed to manipulate platform feed algorithms by pushing specific keywords or hashtags, attempting to boost trending topics, and making content appear more popular than it actually is [50, 57]. We refer to these orchestrated efforts as *web domain promotion* and *content amplification*.

*4.1.1 Detection of Web Domain Promotion.* To identify orchestrated campaigns promoting specific web domains, we examine user similarities based on the URLs they share in their social media posts. To ensure high-quality data and reduce noise, we applied several preprocessing steps. First, we imposed a minimum activity threshold, requiring each user to have shared at least 10 unique URLs. This criterion is consistent with prior work [35, 50] and ensures that the analysis focuses on users with substantial contributions, filtering out those with minimal engagement that could distort the detection of coordinated accounts. Next, we expanded shortened or obfuscated URLs using a URL expansion library[2].

**Co-URL similarity network.** We constructed a bipartite user-URL network, where users and URLs are connected based on the URLs that users share. In the adjacency matrix of the bipartite graph, users are represented by the rows, whereas the columns represent the URLs. In accordance with previous work [35, 50], we applied the Term Frequency-Inverse Document Frequency (TF-IDF) transformation to represent the user-URL matrix. To avoid bias towards overly frequent URLs, we set a maximum document frequency (max DF) of the 90th percentile and a minimum document frequency

---

[1]We will use the term *accounts* to also refer to Facebook pages, Facebook groups, and Telegram channels.

[2]https://github.com/dfreelon/unspooler

(min DF) threshold of 5 occurrences per URL. This ensures that both very rare and overly common URLs are excluded from the analysis, leaving us with a meaningful set of URLs that can identify CoIAs.

The similarity between users is built by comparing their pairwise co-shared URL patterns. This is obtained by projecting the bipartite network into a user-to-user similarity network. For network construction, we computed exhaustive pairwise cosine similarity between user vectors of the bipartite graph. This similarity measure captures how similar users are based on the URLs they shared, with higher similarity scores indicating stronger coordination. This co-URL similarity network forms the foundation to identify coordinated users potentially driving campaigns aimed at redirecting traffic to specific web domains.

**Intra-platform and cross-platform detection.** Being agnostic to the platform under scrutiny, URLs serve as common entities that can be analyzed across different platforms to identify potential cross-platform campaigns. Therefore, we constructed two types of co-URL networks: intra-platform and cross-platform. For the intra-platform network, we computed cosine similarity between all pairs of users within the same platform. For the cross-platform network, we calculated pairwise similarity only between users from different platforms, linking users based on their shared URL patterns.

**Identifying networks of coordinated accounts.** To identify online coordination, we build upon and integrate state-of-the-art strategies that filter either low-weight edges or peripheral nodes from the similarity network to detect CoIAs. The first approach [50] prioritizes the strength of similarity, filtering out low-weight edges likely representing spurious similarities. The second approach [35] focuses on the breadth of similarities, pruning nodes based on their centrality. Luceri et al. [35] demonstrated that eigenvector centrality is an effective network property for identifying the most suspicious users in a similarity network, where coordinated accounts tend to share numerous similarities with other highly connected nodes. The assumption is that coordinated activity typically involves multiple accounts. In a similarity network, coordinated actions manifest as a pronounced collective similarity, where a coordinated account is highly connected (i.e., similar) to numerous other nodes that are themselves well-connected. Consequently, eigenvector centrality has proven to be an effective network property for detecting online coordination [35]. Here, we extend this notion of collective similarity by considering the density of the similarity network. Network density is a measure that quantifies how close a network is to being fully connected and, in this scenario, it provides an exact measure of collective similarity.

In particular, we use network density as a variable to modulate the combination of thresholds for both node and edge filtering. By integrating these strategies, we aim to leverage the strengths of both filtering techniques, constructing a model that account for both the breadth and strength of similarities. This integrated approach is particularly important, as these techniques have not yet been applied to platforms like Telegram and Facebook, where coordination dynamics may vary significantly.

**Density-based network dismantling.** To combine these strategies in a fully unsupervised manner, we developed a novel method for conducting a grid search of parameters in the two-dimensional space defined by node centrality and edge weight distributions.

Specifically, the grid search explores two parameters—the quantile of node centrality and the quantile of edge similarity—and evaluates network density for each combination and every connected component in the similarity graph. We hypothesize that, as we progressively filter the similarity graph by removing edges and nodes, a transitional phase in the density of the connected components will signal potentially coordinated accounts. To ensure robustness, we adopt a conservative approach by evaluating the transitional phase of the smallest density among all connected components in the filtered similarity graph. By controlling the component with the lowest density, we ensure that all other components exhibit higher densities. This conservative choice is driven by the unsupervised nature of the task, prioritizing the minimization of false positives, i.e., legitimate users misclassified as coordinated. Therefore, filtering thresholds were chosen based on the transitional phase of the smallest density of a connected component of the similarity graph, thus, focusing on identifying highly suspicious CoIA. Details of the parameters used for detection are provided in the Appendix.

*4.1.2 Detection of Content Amplification.* Coordinated actors may employ a variety of tactics to achieve their goals. A common tactic is to artificially amplify content on specific topics to create the appearance of widespread grassroots support and manipulate platforms' feed algorithms [38, 49, 57]. To uncover content amplification, we construct a Text Similarity Network (TSN), where nodes represent users linked by the similarity of the content they share. This TSN is then employed to identify a subset of users exhibiting suspiciously coordinated behavior.

The initial step in constructing the TSN involves preprocessing the raw data to ensure the results are meaningful. In line with standard practices [35], we exclude retweets and remove punctuation, stopwords, emojis, URLs, as well as any content with fewer than four words. To capture the semantic nuances of the content, we embed all text data using the SentenceTransformer model *stsb-xlm-r-multilingual*[3]. We then calculate the average cosine similarity between pairs of users, utilizing the highly effective FAISS library[4]. To maintain temporal coherence in the analysis, we assess similarities using one-day sliding windows. The TSN connects users when they post at least one pair of similar tweets, with the average text similarity used to weight the edges between them.

To identify coordinated accounts, we apply thresholds commonly used in the literature [35, 50]. Specifically, we filter out any edge with a similarity below 0.95 and classify users as coordinated if their nodes rank within the top 0.5% for eigenvector centrality.

## 4.2 Characterizing content pushed by CoIA

In this section, we present an overview of the methods used to characterize textual content amplified by coordinated actors across various social media networks. This characterization encompasses topic analysis, AI-generated content detection, and an assessment of content credibility.

*4.2.1 Topic Analysis.* Coordinated accounts often amplify specific narratives or themes to steer public attention toward polarized, inflammatory, or misleading discussions. To uncover the agendas

these accounts seek to promote, we use BERTopic [20], a state-of-the-art tool for topic extraction. For a detailed description of BERTopic's methodology, we refer readers to [20]. This approach also helps identify patterns in shared content, improving our understanding of coordinated activities and their intended impact on public discourse.

Once coordinated accounts are identified using the methodology outlined in §4.1.2, we apply BERTopic to the entire content corpus. This allows us to map the topics promoted by coordinated accounts and compare them with those shared by organic users (i.e., users not classified as coordinated). We choose BERTopic over alternatives such as LDA or GPT-based approaches because it offers an effective balance between accuracy and scalability.

*4.2.2 AI-generated content detection.* To identify AI-generated textual content, we employ the model from [11]. Originally designed only for tweets, we extend its applicability to other platforms by developing a new validation set of approximately 2,000 samples, tailored to ensure robust multi-platform detection. This set was built using Llama 3.1 3B Instruct[5] and GPT-4o[6], along with older social media datasets from Telegram, Twitter, and Facebook (~2010-2015). We assume that these older datasets do not contain any AI-generated content. To complement this authentic content with AI-generated text, we prompt LLMs to generate new content that matches the original texts' characteristics, ensuring a similar distribution of topics and lengths. This approach ensures the AI-generated texts closely mirror the non-AI content, providing a robust validation process. Favoring a conservative approach, we prioritized precision over recall, achieving precision values ranging from 0.87 to 0.97 in the detection of AI-generated content in the validation set. The complete set of results is presented in the Appendix.

*4.2.3 Credibility Assessment.* We assess the credibility of web domains using Media Bias/Fact Check (MBFC). MBFC is an independent watchdog that rates news outlets on a 6-point factuality scale, ranging from Very Low to Very High. For each post in our datasets, we systematically extract, expand, and parse all embedded URLs, checking whether the URL belongs to a low- or high-credibility domain from our lists.

Given recent news[7][8] and prior instances of interference by Russian agencies in U.S. elections [4], we also examined the potential amplification of Russian media outlets. Following a similar approach to previous work [53], we utilize the VoynaSlov dataset [51] to obtain a list of 23 state-affiliated Russian websites. We then check whether the extracted URLs link to one of these web domains.

Finally, we employ a similar approach to identify content linked to conspiracies, such as QAnon, given the relevance of fringe theories during the 2020 US Election, and its aftermath [54, 57]. Building on the methodology of [58], we detect posts sharing QAnon content by utilizing a list of keywords commonly associated with the conspiracy, as outlined by [54].

---

[3]https://huggingface.co/sentence-transformers/stsb-xlm-r-multilingual
[4]https://github.com/facebookresearch/faiss/

[5]https://huggingface.co/meta-llama/Meta-Llama-3-8B-Instruct
[6]https://openai.com/index/hello-gpt-4o/
[7]https://www.bbc.com/news/articles/c8rx28v1vpro
[8]https://www.state.gov/u-s-department-of-state-takes-actions-to-counter-russian-influence-and-interference-in-u-s-elections/

**Figure 1: Cross-platform coordination network showing user coordination across four social media networks.**

| Domain | Shares | Factuality | Leaning |
|---|---|---|---|
| ruptly.tv | 2,117 | Mixed | RIGHT-CENTER |
| rt.com | 1,941 | Very Low | RIGHT-CENTER |
| odysee.com | 1,602 | Low | RIGHT CONSPIRACY |
| dailymail.co.uk | 1,159 | Low | RIGHT |
| thegatewaypundit.com | 746 | Very Low | EXTREME RIGHT |

**Table 2: Top-5 domains shared in Telegram by coordinated accounts. Columns represent, from left to right: domain, the number of shares among coordinated accounts, and the factuality and political leaning scores based on data from MBFC.**

| Domain | Shares | Factuality | Leaning |
|---|---|---|---|
| foxnews.com | 880 | Mixed | RIGHT |
| foxbusiness.com | 13 | Mixed | RIGHT-CENTER |
| newsbreakapp.com | 5 | NA | NA |
| go.shr.lc | 2 | NA | NA |
| lifenews.com | 2 | Low | FAR RIGHT |

**Table 3: Top-5 domains shared in Twitter by coordinated accounts. Columns represent, from left to right: domain, the number of shares among coordinated accounts, and the factuality and political leaning scores based on data from MBFC.**

## 5 RESULTS

This section presents the results obtained from the detection of online coordination. We first present the results for the detection of *web domain promotion* and *content amplification*. Following the identification of these campaigns, we examine the content pushed by coordinated networks and the engagement received by suspicious actors focusing on several interaction and engagement metrics.

### 5.1 Detecting CoIA for Web Domain Promotion (RQ1)

To detect CoIAs that push web domains within and across platforms, we constructed i) co-URL similarity networks for each platform separately. Filtering out non-suspicious users (see §4.1.1), we then identified clusters of *intra-platform coordinated accounts* within each platform; ii) a *inter-platform* co-URL similarity network considering similarities only between accounts from different platforms. We then detected *inter-platform coordinated accounts* as described in §4.1.1. To have a comprehensive overview of intra- and inter-platform coordinated networks, we created a *cross-platform network* by combining the coordinated networks into a single, unified network, where nodes are connected if linked in either the intra- or inter-platform coordinated network. The primary result of this analysis is displayed in Figure 1, which portrays coordination within and across the three platforms under observation.

*5.1.1 Intra-Platform Coordination.* We first analyze each intra-platform CoIA based on the co-URL similarity networks extracted from each platform separately. We describe intra-platform CoIAs as follows.

**Telegram:** We identified 33 highly coordinated channels co-sharing URLs to web domains with a partisan slant and low factuality. As shown in Table D.2, the most frequently shared domains are predominantly right-leaning and of low credibility, as assessed by MBFC[9]. Notably, the table highlights the presence of a Russian state-controlled media outlet (RT.com) and its video-on-demand subsidiary (Ruptly.tv), which could indicate a potential foreign effort to interfere in the election. Both RT.com and Ruptly.tv have previously been accused of orchestrating campaigns to influence U.S. elections via social media[10].

Additionally, this CoIA promotes a far-right website (thegatewaypundit.com) and an extremist-friendly video platform (odysee.com), suggesting ties to fringe ideas. A manual review of these Telegram channels, including their profile descriptions (see Table D.2) and shared messages (see Table D.6), reveals a strong prevalence of content and accounts promoting conspiracy theories, particularly those related to COVID vaccines, 5G, and alternative news media.

𝕏 **(Twitter):** We identified a network of 19 coordinated accounts predominantly promoting content from right-leaning domains, as listed in Table 3. A manual review of this Twitter CoIA reveals that many of these coordinated accounts share similar profile descriptions, reflecting narratives tied to religious and conservative principles (see Table D.3). Additionally, an analysis of the messages (see Table D.7) shows a strong presence of politically partisan content, predominantly aligned with right-leaning ideologies.

**Facebook:** On Facebook, no well-known media outlets dominate the shared content. However, the most shared domain is a partisan news outlet, as suggested by the semantics of its title:

---
[9]https://mediabiasfactcheck.com/
[10]https://www.nbcnews.com/politics/2020-election/facebook-blocks-russia-backed-accounts-other-sites-keep-churning-out-n1242683

gorightnews.com. Additionally, there is a direct connection to a public activist known for political campaigns[11] and their website: peterboykin.com.

The related campaigns and Facebook accounts primarily focus on the "Gays for Trump" movement, an American LGBTQ organization that advocates for former U.S. President Donald Trump and his administration [12].

The top-engagement messages (see Table D.8) and bios (see Table ??) indicate strong support for right-leaning ideologies, consistent with the stated aims of this CoIA.

*5.1.2 Cross-Platform Coordination.* The analysis of the cross-platform network reveals distinct patterns. The unified cross-platform network, displayed in Fig. 1, highlights a giant component connecting coordinated accounts from both Telegram and Twitter, while the Facebook coordination network remains largely disconnected. A few noteworthy observations:

First, we observe that most cross-platform connections occur between Twitter accounts and Telegram channels, likely due to their higher representation in the dataset.

Examining the bios of the top users by degree, as shown in Table D.9, we note a prominent presence of non-mainstream news outlets and conspiracy theories, such as the flat earth one.

Second, when focusing on the bridge nodes between the Telegram-coordinated cluster and the cross-platform giant component, we find highly influential channels with up to 24,000 subscribers. These channels often feature bios referencing free speech and religion:

- "The FIGHT for FREEDOM - :Aron TRUTH Social"
- "Q Reee-searchers Watchers"
- "'Arise and shine, for your light has come, and the glory of the LORD rises upon you.' Isa. 60:1 WE RISE beyond the challenges of yesterday to become better stewards of tomorrow!"

Similarly, the bridge node between the Twitter-coordinated cluster and the cross-platform giant component is a highly influential account with 32,000 followers. The bio reads: "NO DM'S. Beautiful disaster. Self-proclaimed arbiter of great ideas. Here to annoy the dumb asses. NO LISTS!! #Imvotingforafelon #animallover", indicating a strong partisan flavor.

Finally, the largest component of the cross-platform coordination network is dominated by a mix of Telegram and Twitter accounts promoting domains such as `magapac.com`, QAnon-related narratives (e.g., "WWG1WGA"[13]), and accounts with partisan bios like:

- "We The Ultra Patriots is made up of patriotic Americans dedicated to exposing crimes against humanity, false flags, lies, and corruption. We The Ultra Patriots are..."

A closer look at the top shared domains within the coordinated network (see Table 4) reveals that most are partisan, with the exception of `truthsocial.com`, an alternative non-mainstream social media platform[14]. Additionally, seven of the shared domains are Russian state-affiliated websites[15], including `tv5`, `rbc`, `gazeta`, `ruptly`, and `mil`, further suggesting potential Russian information

---

[11]https://en.wikipedia.org/wiki/Peter_Boykin
[12]https://en.wikipedia.org/wiki/Gays_for_Trump
[13]https://en.wiktionary.org/wiki/WWG1WGA
[14]https://en.wikipedia.org/wiki/Truth_Social
[15]https://github.com/chan0park/VoynaSlov/tree/master

| Domain | Shares | Factuality | Leaning |
|---|---|---|---|
| thegatewaypundit.com | 23,513 | Very Low | EXTREME RIGHT |
| zerohedge.com | 8,331 | Low | RIGHT CONSPIRACY |
| truthsocial.com | 7,800 | NA | NA |
| nypost.com | 6,797 | Mixed | RIGHT-CENTER |
| theepochtimes.com | 6,241 | Mixed | RIGHT |

**Table 4: Top domains shared by cross-platform coordinated accounts. Columns represent, from left to right: domain, the number of shares among coordinated accounts, and the factuality and political leaning scores based on data from MBFC.**

operations to interfere with the election discourse. Examining the posts and messages with the highest engagement (see Table D.9 in the Appendix), we observe the prevalence of ultra-MAGA narratives, conservative religious themes, and environmental news.

## 5.2 Detecting CoIA for Content Amplification (RQ2)

In this section, we present our findings on the specific topics amplified by coordinated users in the context of the 2024 U.S. Election. Using BERTopic, as outlined in §4.2.1, we identify the themes these users aim to promote and compare them to those emerging organically within each platform. We construct a *Text Similarity Network* by connecting users with similar content, filter out edges below a similarity threshold of 0.95, and identify coordinated users as those in the top 0.5% by eigenvector centrality. For further details, the reader can refer to §4.1.2.

Next, we train a BERTopic model on the entire content corpus, allowing it to automatically identify topics and assign each piece of content to a specific topic. We then categorize the content based on the user who shared it, distinguishing between coordinated and organic users (i.e., those not identified as coordinated). For each platform, we report the top-10 most prevalent topics within the coordinated cohort and compare their prevalence to that within the organic population. This approach helps us identify the themes promoted by coordinated users, with larger differences between the two groups indicating topics that are more characteristic of coordinated activity. In the remainder of this section, we present our findings on coordination-driven content amplification for each of the three platforms analyzed.

**Telegram:** Based on the significantly high similarity of their shared content, we identified 57 coordinated Telegram channels Among these channels, public health is a prominent topic of discussion, as can be seen in Figure 2, with skepticism around the COVID-19 vaccine emerging as the most frequently discussed theme. Additionally, many messages promote alternative medicine, and there is notable discussion surrounding a conspiracy theory related to bird flu vaccination. Telegram's coordinated channels also focus on strictly political discussions, particularly on the U.S. presidential debate, the Russia-Ukraine conflict, and Donald Trump's legal challenges. Interestingly, we observe discourse around immigration issues in Ireland, which appears to be echoed by accounts associated with the MAGA movement. Notably, three out of the ten topics involve conspiracy theories related to flat-earth beliefs, bird flu, and

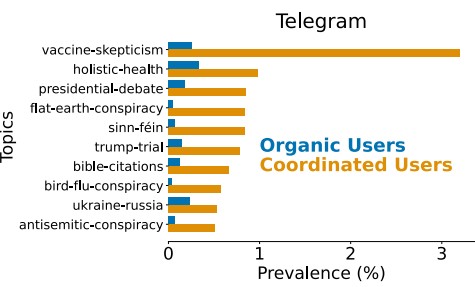

Figure 2: Content analysis of coordinated channels. We report the top-10 most recurring topics and we compare their prevalence against the organic Telegram discourse.

| Keyword | Count / Prevalence | |
| --- | --- | --- |
| | Coordinated | Organic |
| wwg1wga | 1260 (0.59%) | 520 (0.05%) |
| plandemic | 619 (0.29%) | 102 (0.01%) |
| adrenochrome | 448 (0.21%) | 118 (0.01%) |
| qanon | 260 (0.12%) | 107 (0.01%) |
| deepstate | 193 (0.09%) | 87 (0.008%) |

Table 5: Keyword statistics in coordinated and organic content related to QAnon conspiracies on Telegram.

antisemitism. The most representative content for each topic, as identified by BERTopic, is provided in the Appendix.

We also assess the presence of QAnon-related keywords in content shared by both coordinated and organic channels. Approximately 1.66% of content from coordinated users contains at least one QAnon-related keyword, compared to only 0.12% of content from the organic population. This substantial difference is evident in Table 5, where we list the top five keywords present in the coordinated group. Although coordinated channels make up only 0.36% of the total Telegram dataset, they show a marked prominence, both in absolute and relative terms, in sharing QAnon-related keywords compared to the organic channels. These findings indicate a propensity for Telegram CoIAs to disseminate conspiracies.

**Twitter:** On Twitter, we identify 221 coordinated users based on the extracted *Text Similarity Network*. Figure 3 displays the top 10 topics shared by these coordinated accounts, all of which pertain to social, economic, or political issues. The most prevalent topic — and one that diverges significantly from those shared by the organic group — focuses on the Independent candidate Robert F. Kennedy, with a subset of coordinated accounts promoting him as a serious contender in the U.S. presidential election. Coordinated users also concentrate on the Democratic Primary race in Westchester County (labeled as 'bowman-lattimer' in the figure), where the most representative tweets endorse candidate Lattimer. This discussion appears to be framed around the anti-Israel stance of the incumbent, Bowman. The remaining topics revolve around polarizing issues such as COVID-19 vaccines, U.S. border security, and various scandals, including the Epstein files and Hunter Biden.

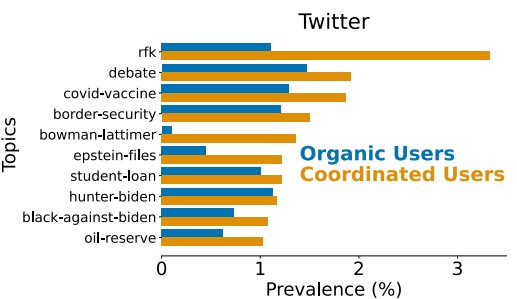

Figure 3: Content analysis of coordinated users. We report the top-10 most recurring topics and we compare their prevalence against the organic Twitter discourse.

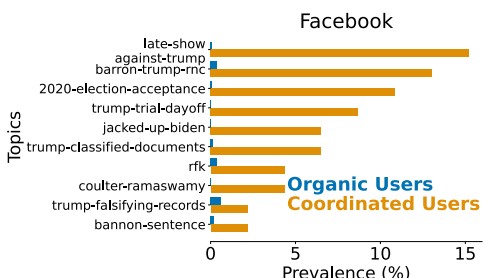

Figure 4: Content analysis of coordinated users. We report the top-10 most recurring topics and we compare their prevalence against the organic Facebook discourse.

Additionally, these users discuss perceived opposition within the Black community to President Biden, critiques of the student loan forgiveness policy, and concerns over the Biden administration's sale of oil reserves. No conspiracy theories were detected among the coordinated users, and only a minimal amount of QAnon-related content was found within the organic group. The most frequent keyword, 'qanon,' appeared just five times.

**Facebook:** We identify 16 coordinated users in the Facebook dataset. As shown in Figure 4, most topics promoted by these users are left-leaning. A manual inspection reveals that all 16 accounts are associated with The Young Turks network[16], which Media-Bias classifies as left-leaning[17]. Apart from a news item quoting a GOP representative accusing Biden of using 'supplements' (i.e., 'jacked-up Biden'), we found no trace of misleading, conspiracy or inflammatory content.

## 5.3 Generated Engagement and AIGC

To comprehensively assess the impact of coordination strategies in the 2024 U.S. election online debate, we analyze the level of engagement generated by coordinated actors and the extent of their reliance on AI-Generated Content (AIGC). For each platform, we begin by merging the sets of coordinated actors identified through co-URL sharing (web domain promotion) and content similarity

---

[16]https://en.wikipedia.org/wiki/The_Young_Turks
[17]https://www.allsides.com/news-source/young-turks-media-bias

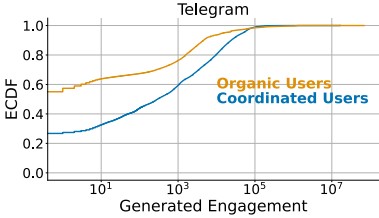 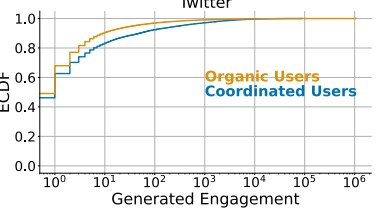 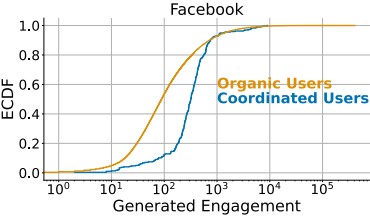

**Table 6: Empirical Cumulative Distribution of the generated engagement by coordinated and organic users.**

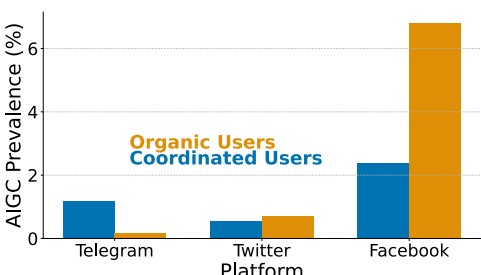

**Figure 5: Distribution of AIGC shared by coordinated and organic users across the three different platforms.**

patterns (content amplification). We observe that the intersection between these two patterns is modest, with overlap rates of 1.3%, 0.5%, and 0% for Telegram, Twitter, and Facebook, respectively. This is in line with previous findings showing that different strategies are used to push diverse agendas through distinct coordinated networks [35, 42]. Aggregating coordinated actors from distinct CoIA networks, we obtain 233 coordinated actors on Telegram, 764 on Twitter, and 25 on Facebook.

Table 6 shows the distribution of engagement generated by coordinated and organic users across all platforms. In each case, coordinated users tend to generate less engagement than their organic counterparts. This is expected, as the organic group includes major media outlets and influencers that significantly boost overall engagement. However, it is noteworthy that the engagement from coordinated actors is substantial, and particularly on Twitter and Facebook, the engagement distributions between coordinated and organic users are quite similar.

We also assess the prevalence of AIGC that was shared across the three platforms by both coordinated and organic users. These results are shown in Figure 5. Facebook is the social media platform where most AIGC is produced by both coordinated and organic accounts. While the AIGC prevalence is limited, we observe a striking difference in the activity between coordinated and organic users on Telegram and Facebook, whereas it is quite balanced on Twitter. Interestingly, AI-generated content is predominantly diffused by coordinated actors on Telegram, while an opposite trend is observed on Facebook.

## 6 CONCLUSIONS

This paper introduces a novel, network-based framework for detecting coordinated inauthentic activity (CoIA) across multiple social media platforms. Unlike traditional methods that focus on isolated platform-specific analyses, our approach emphasizes cross-platform behaviors by constructing similarity networks that capture community-wide patterns of content sharing. By prioritizing both intra- and cross-platform connections, we are able to detect subtle yet significant coordination signals that remain hidden when platforms are analyzed in isolation.

Our unsupervised, network-based methodology allows us to identify a range of influence campaigns aimed at directing traffic towards specific narratives and domains. These campaigns frequently promote content that is partisan, low-credibility, and conspiratorial in nature. The model not only detects domestic actors but also reveals the systematic promotion of foreign-affiliated media, particularly Russian state-sponsored outlets, across platforms like Telegram and $\mathbb{X}$. This evidence points to a coordinated effort to amplify certain messages and domains across distinct platforms, particularly in the context of the 2024 U.S. Election, underscoring the necessity of a unified, cross-platform approach to tackle such influence operations.

*Limitations.* Despite its strengths, our study has certain limitations. First, while the model draws on data from Telegram, Facebook, and $\mathbb{X}$, the inherent biases in data collection, as well as platform-specific behaviors, could impact the generalizability of our results. Additionally, our framework merges networks with different similarity distributions, making the detection of coordination inherently relative to each dataset. This relative approach does not provide an absolute measure of coordination strength across platforms, which could limit the comparability of results.

Future work should address these limitations by incorporating a null model to evaluate the statistical significance of coordination patterns. This would allow for a more robust interpretation of the results and ensure that the identified coordination signals are not artifacts of the underlying data distributions. Expanding the framework to include additional platforms and exploring a wider range of behavioral signals would further enhance its ability to detect diverse forms of coordinated inauthentic activity.

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

| Platform | Precision | Recall |
|----------|-----------|--------|
| General | 0.9064 | 0.5927 |
| Facebook | 0.8723 | 0.6686 |
| Twitter | 0.8935 | 0.5613 |
| Telegram | 0.9598 | 0.5956 |
| Reddit | 0.895 | 0.5368 |
| Additional set | 0.9697 | 0.5 |

**Table B.1: Precision and Recall for each platform**

## A    DATASET COLLECTION

**Keywords.** Joe Biden, Donald Trump, 2024 US Elections, US Elections, 2024 Elections, 2024 Presidential Elections, Biden, Joe Biden, Joseph Biden, Biden2024, Donald Trump, Trump2024, trumpsupporters, trumptrain, republicansoftiktok, conservative, MAGA, KAG, GOP, CPAC, Nikki Haley, Ron DeSantis , RNC, democratsoftiktok, thedemocrats, DNC, Kamala Harris, Marianne Williamson, Dean Phillips, williamson2024, phillips2024, Democratic party, Republican party, Third Party, Green Party, Independent Party, No Labels, RFK Jr, Roberty F. Kennedy Jr. , Jill Stein, Cornel West, ultramaga, voteblue2024, letsgobrandon, bidenharris2024, makeamericagreatagain, Vivek Ramaswamy, JD Vance, Assassination, Tim Walz, WWG1WGA.

**Additional dataset statistics.** We present the distribution of the number of posts per user and post's length for each platform.

## B    AI DETECTION CLASSIFIER VALIDATION

We utilized the AI-generated text detection classifier from [11], originally trained to identify AI-generated content in tweets. To extend the applicability of this classifier to other platforms, we constructed an external validation set using diverse, older datasets. Specifically, we built upon four datasets: `Twitter-2010`[18], `Facebook`[19], and `Telegram`.

From each of these datasets, we sampled 500 texts ranging in length from 125 to 1000 characters. Given the release years of these datasets, we assume that all texts are non-AI generated. To create the AI-generated class, we used GPT-4o and Llama 3.1 3B Instruct, prompting them to generate new texts with similar topics and lengths as the original non-AI texts. We performed this generation for each non-AI text and sampled 250 AI-generated texts from GPT and 250 from Llama.

Thus, for each platform, the validation dataset consists of 500 non-AI generated texts paired with 500 AI-generated texts: 250 from GPT and 250 from Llama, with both AI-generated sets matching the original texts in topic and length.

To further enhance the generalizability of the validation dataset, we incorporated non-AI generated content from additional sources: Reddit comments[20], IMDB reviews[21], movies corpus[22], and Yelp 2013[23]. AI-generated content for these datasets was sourced from tweetHunter[24] and GPT, which was prompted to generate texts on specific topics such as American football, climate change, the Soccer World Cup, the Gaza conflict, U.S. politics, and vaccines.

The validation results are displayed in Table B.1.

---

[18]https://archive.org/details/twitter_cikm_2010
[19]https://www.kaggle.com/datasets/sheenabatra/facebook-data
[20]https://zissou.infosci.cornell.edu/convokit/datasets/reddit-coarse-discourse-corpus/
[21]https://ai.stanford.edu/~amaas/data/sentiment/
[22]https://zissou.infosci.cornell.edu/convokit/datasets/movie-corpus/
[23]https://www.yelp.com/dataset/download
[24]https://tweethunter.io/

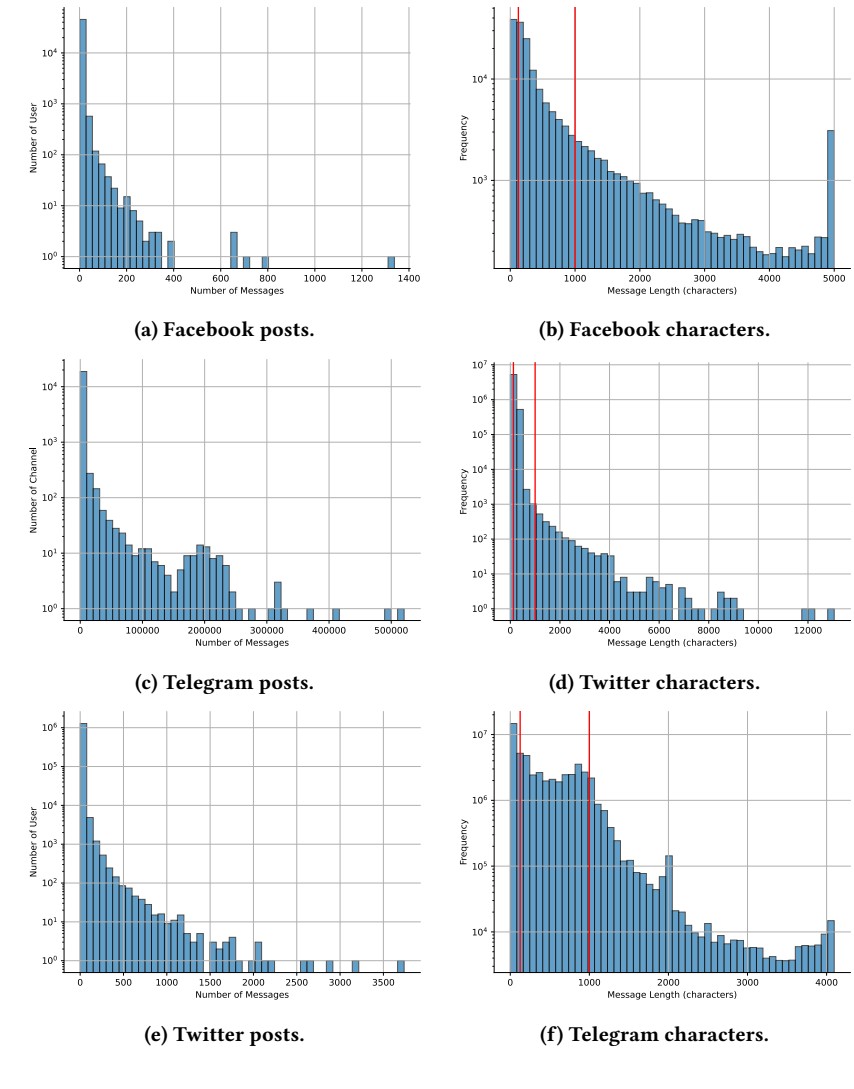

(a) Facebook posts.

(b) Facebook characters.

(c) Telegram posts.

(d) Twitter characters.

(e) Twitter posts.

(f) Telegram characters.

Figure .1: Distributions of posts and characters across platforms.

| Bio |
| --- |
| CV Lies & 5G Dangers - Discussions |
| Scammers may try to impersonate Real World News Channel. Don't trust any messages you get from them. They are not us. Block and report them immediately. |
| Covering News, Military information, across Wiltshire & the Southwest areas and for connecting people together uk_flag |
| Credence - Breaking News |
| Truth. Faith. Freedom. |

**Table D.2: Current bios of top-5 coordinated channels in Telegram sorted by degree.**

| Bio |
| --- |
| Have integrity. Be Just. Precinct Chairman. Christian. Pro-Life. Conservative Political Activist. #CruzCrew |
| NO DM'S. Beautiful disaster. Self proclaimed arbiter of great ideas. Here to annoy the dumb asses. NO LIST!! #Imvotingforafelon #animallover |
| CHRISTIAN, ULTRA MAGA, PATRIOT, CONSERVATIVE, 2A, FJB, TRUMP WON, TRUMP2016, TRUMP2020, TRUMP2024, ELON MUSK, YELLOWSTONE, DALLAS COWBOYS |
| just a opinionated old cowgirl from WY Traditional work ethic, traditional values |
| studied spirituality under a minister study 15 years worked 30+years. I write as I see them, past present and maybe future. don't expect you to agree but think |

**Table D.3: Current bios of top-5 coordinated accounts in Twitter sorted by degree.**

## C GRID SEARCH FOR SIMILARITY AND CENTRALITY THRESHOLDS

To identify coordinated accounts in the co-URL similarity network, we filter edges and nodes based on cosine similarity and eigenvector centrality. Two nodes are connected if they co-share URLs, and the strength of these connections is represented by the cosine similarity between their TF-IDF vectors across the space of unique URLs.

We employ two filtering techniques commonly found in the literature: edge filtering, based on cosine similarity, and node filtering, based on eigenvector centrality. The thresholds for these filters are determined by the percentiles of the distributions of edge similarity and node centrality.

Assuming that coordinated accounts manifest as dense components in the similarity graph, we use the density of each connected component as a key indicator of coordination. To ensure robustness, we adopt a conservative approach, using the minimum density across all connected components after filtering as a comprehensive quality measure of the graph.

Figure C.3 presents the results of our grid search across these parameters. The x-axis represents the percentile of edge similarity, while the y-axis corresponds to the percentile of eigenvector centrality. The z-axis indicates the minimum density of the similarity graph after filtering based on the selected x-y percentiles.

For each platform, we identified the optimal thresholds by observing sharp changes in minimum density and high overall density values. These threshold combinations define the coordinated accounts on each platform. Specifically:

- For Facebook, we selected (50%, 45%) as the minimum density increased from 0.73 and 0.80 to 0.90.
- For Twitter, we chose (85%, 99%), consistent with previous studies, leading to an increase from 0.74 to 0.99.
- For Reddit, we selected (85%, 80%), where the minimum density shifted from 0.81 and 0.89 to 0.96.
- For Telegram, we used (99%, 99%), focusing on a significant jump from 0.18 to 0.87.

Other threshold combinations were evaluated qualitatively, yielding similar results. More quantitative methods for threshold selection are left for future work.

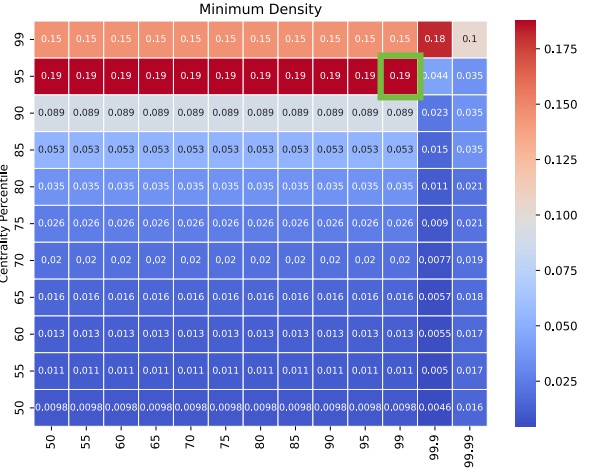

**Figure C.2: Thresholds grid search for filtering the Cross-platform similarity graph: x-axis edge similarity quantile, y-axis node centrality quantile. z-axis is the minimum graph density across all connected components of the filtered co-url similarity graph. The green square corresponds to the selected thresholds.**

## D CHARACTERIZATION OF COORDINATED USERS

We report the bios of the top 5 coordinated accounts by degree, at the time of writing, and the top 5 messages by total engagement written by coordinated accounts in each platform.

## E TOPIC ANALYSIS

| Top-5 Messages |
|---|
| "US Secretary of State Antony Blinken Walking the Congressional Office Building Halls Surrounded by Police 'War criminal', 'genocide secretary' Protestors reminding him of his complicity in crimes against humanity. RealWorldNewsChannel RealWorldNewsChat" |
| "Its sickening, disgusting money laundering MF's. In a meantime the people of United States are suffering! Can't afford basic necessities!!" |
| "I like how zoinists shout poor me when they get caught out for there evil crimes Normal Jews hate zoinist Jews People of the world don't hate Jews People of the world hate zoinists Jews n zoinists in general n what they stand for They think there the chosen ones and that gives them the rite to kill n genecide inercent people Zoinisium is hated by all races of the world And has no place for peace in humanity " |
| "Ukrainians in Galway are being advised to vote for two Nigerians and a Labour candidate to best serve their interests. 'Under no circumstances vote for radicals - Irish Freedom Party, Independent Ireland, The Irish People and all others who have the slogans 'Ireland for the Irish'. #ForeignInterference " |
| "Former CNN Anchor Chris Cuomo Admits to Suffering from a COVID Vaccine Injury ICYMI: There's been a major shift in the official narrative. Follow Vigilant_News " |

**Table D.6: Top-5 messages by total engagement for coordinated channels in Telegram**

| Top-5 Posts |
|---|
| ""WE THE PEOPLE" don't play by your dictator rules. Just answer the damn question, crybaby. White House correspondents fire back after Biden snaps at reporter for refusing to 'play by the rules'" |
| "If Biden wins there will no more arguing because the will not be anything left to argue about. He's handing the keys over to illegals and the new world order. In the end, the joke will be on the democrats. The WHO/WEF hates them too and knows they're imbeciles. Everyone knows" |
| "'Shameful': GOP lawmaker shreds 'AWOL' Biden for throwing Jews 'under the bus' amid anti-Israel protests If this president, so-called president doesn't personify evil destruction division of this country, I don't know what or who would !" |
| "Biden DHS docs suggested Trump supporters, military and religious people are likely violent terror threats. HaHaHa! I guess we also have Santa Claus/Easter bunny over here in training. Give me a break, we are trying to save this country from the bad guys!" |
| "DeSantis spox dunks on NYT 'fact-check' on terrorists entering southern border: 'Awaiting your correction' They've been entering for last 3 years. Biden admin has no damn idea who is coming in. Even ones they think they know are using others' identity" |

**Table D.7: Top-5 posts by total engagement for coordinated accounts in Twitter**

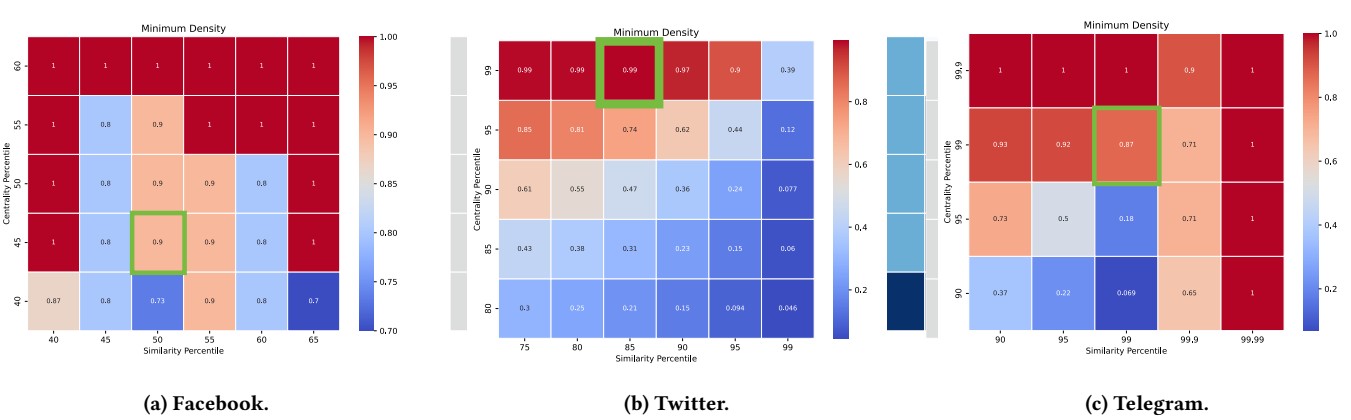

(a) Facebook.                                           (b) Twitter.                                           (c) Telegram.

**Figure C.3: Thresholds grid search for filtering similarity graph: x-axis edge similarity quantile, y-axis node centrality quantile. z-axis is the minimum graph density across all connected components of the filtered co-url similarity graph. The green square corresponds to the selected thresholds.**

**Top-2 Posts**

"Youth Vote Shifts Toward Trump in 2024 Election https://gorightnews.com/youth-vote-shifts-toward-trump-in-2024-election/ Maybe the kids are alright... #GoRightNews Recent polls indicate a surprising trend: young voters are warming to Donald Trump in the 2024 presidential election. The question arises: Are these voters aligning with Trump's policies, or is President Biden driving them away? The answer might lie in a combination of both factors. Polling Data Reveals Shift According to the latest New York Times poll, young voters aged 18 to 29 favor Biden by a slim margin of two points, 47% to 45%. A Quinnipiac poll shows Trump leading Biden among voters aged 18 to 34, 48% to 47%. This is a stark contrast to the 2020 presidential election, where Joe Biden secured the youth vote by a significant 24%. The last time a Republican won this demographic was in 1988. Biden's Struggling Message Aidan Kohn-Murphy, founder of Gen Z for Change—a group that supported Biden in 2020—stated in the Washington Post, "Biden is out of step with young people on a number of key issues." Key issues where Biden seems to be losing support include: The War in Gaza: A majority of young voters, 51%, support the Palestinians, while only 15% support Israel. TikTok Ban: Biden's support for a TikTok ban is perceived as an attack on free speech. Consequently, 67% of Gen Z voters say this makes them less likely to vote for him. Economic Challenges: High inflation and interest rates have made essential costs like food and housing unaffordable for young people entering the workforce or trying to purchase their first home. Trump's Resonating Message The 45th president's message appears to be resonating with young voters for several reasons: Gaza Conflict: While Trump supports Israel, he promises to bring a peaceful end to the conflict, citing his administration's four years of peace as evidence of his capability. TikTok Engagement: Trump recently joined TikTok, emphasizing that Biden wants to shut the platform down, thus appealing to younger users. Economic Performance: When asked about Trump's handling of the economy, 65% of young voters approved, compared to just 33% for Biden. Social Media Dynamics According to CredoIQ, a social media analytics firm, nearly 25% of the top left-leaning content creators on TikTok have posted anti-Biden content in the first four months of 2024, garnering over 100 million views. This content is often created by young, non-white liberals who share the belief that the U.S. Government, and specifically Joe Biden, aims to restrict free speech and information flow. Trump's Adaptation to New Media In 2016, Trump broke political norms with an 'anyplace, anytime' approach, dominating cable TV. For the 2024 campaign, he has adapted this strategy to new media, appearing on podcasts, YouTube shows, and attending live events such as UFC and Formula 1. He has also made campaign stops at local venues, including bodegas, firehouses, and even in the South Bronx. The shift in the youth vote suggests a significant realignment in political affiliations, one that underscores the importance of addressing the issues most pertinent to young Americans. This analysis highlights the potential impact of these changing dynamics on the upcoming presidential election. [Source: Washington Post, Axios] https://archive.is/kk3HU https://www.axios.com/2024/06/13/trump-election-young-voters-polling #GoRightNews Shared by Peter Boykin - American Political Commentator / Citizen Journalist / Activist / Constitutionalist for Liberty Web: https://PeterBoykin.com Kick: http://Kick.com/PeterBoykin YouTube: https://youtube.com/PeterBoykinForAmerica Twitter: https://twitter.com/GoRightNews Telegram: http://t.me/realpeterboykin Rumble: http://Rumble.com/GoRightNews Like the Content? Please Support! - Go Right News: Stripe: https://gorightnews.com/donations/support-gorightnews/ Cash App: http://cash.app/$PeterBoykin1"

"Rising Costs Highlight Challenges for American Families Cost of rent, energy, and other essentials surged in May In an alarming trend, the cost of essentials such as rent, energy, and groceries continues to surge, underscoring the persistent financial challenges faced by American families. While the overall Consumer Price Index (CPI) showed a slight stabilization, the specifics reveal a stark reality of escalating living expenses in our Constitutional Republic. Analyzing the Numbers The CPI indicated a 3.3% rise in overall inflation compared to the previous year. Although this marks a slight decrease from April's 3.4% and a significant drop from the 9.1% peak in June 2022, it remains well above the Federal Reserve's target rate of 2%. This persistent inflation underscores the ongoing economic strain on American households. Historical Context It's noteworthy that during the four years of Donald Trump's presidency, the average inflation rate was maintained at a modest 1.9%. This comparison highlights a more stable economic period and suggests a need for policies that can effectively manage inflation without compromising the financial well-being of citizens. Essential Expenses on the Rise A closer examination of the May report reveals substantial increases in essential costs: Rent: Up by 5.4% Mortgage: Up by 5.6% Hospital services: Up by 7.2% Car insurance: Up by 20.3% Electricity: Up by 5.9% Ground beef: Up by 4.9% Steak: Up by 5.7% Bacon: Up by 6.9% Hot dogs: Up by 7.3% These increases in essential goods and services strain the budgets of American families, making everyday living increasingly unaffordable. Public Sentiment A recent poll reflects the public's discontent, with only 31% of voters approving of President Biden's handling of inflation, while a significant 61% disapprove. This sentiment underscores the urgent need for effective economic policies that address the real concerns of the populace. Administration's Response The Biden administration continues to assert progress in combating inflation. A statement from the White House on social media claimed: "Today's report shows continued progress in lowering inflation. President Biden knows that costs are still too high for many families and we still have a lot more to do. That's why he will keep fighting to lower drug costs, grocery prices, and energy bills." Critical Perspective However, many Americans find these assurances lacking. The everyday experience at grocery stores and gas stations starkly contrasts with the administration's optimistic declarations, leading to a disconnect between government rhetoric and public reality. As a Constitutional Republic dedicated to ensuring democracy and the well-being of its citizens, it is imperative that our government implements policies that stabilize the economy and reduce the financial burden on American families. The rising costs of essential goods and services are a pressing concern that requires immediate and effective action to safeguard the economic future of our nation. [Source: Poll, Whitehouse on X, BLS] https://d3nkl3psvxxpe9.cloudfront.net/documents/econTabReport_-maqVHQt.pdf https://x.com/WhiteHouse/status/1800957041390792843 https://www.bls.gov/news.release/pdf/cpi.pdf #GoRightNews Shared by Peter Boykin - American Political Commentator / Citizen Journalist / Activist / Constitutionalist for Liberty Web: https://PeterBoykin.com Kick: http://Kick.com/PeterBoykin YouTube: https://youtube.com/PeterBoykinForAmerica Twitter: https://twitter.com/GoRightNews Telegram: http://t.me/realpeterboykin Rumble: http://Rumble.com/GoRightNews Like the Content? Please Support! - Go Right News: Stripe: https://gorightnews.com/donations/support-gorightnews/ Cash App: http://cash.app/$PeterBoykin1"

**Table D.8: Top-2 posts by total engagement for coordinated pages in Facebook**

| Top-5 Messages |
| --- |
| "!You are a Golden Child!
A Golden Child 94 Unicorn 94 Third Eye 94 Covenant 94 Harmony 94 Praise God 94 Divine Gene 94 Nine Six 94 Blue Eyes 94 Indigo Child 94 Ultra Maga 94 John John 94 White Hat 94 American Eagle 94 Carry On 94 New Earth 94 Pineapple 94
PassionForFruit" |
| "#CAWildfireSituationUpdate as of 6-16-2024, 5:03 P.M. PST.
#PostFire - 12,266 acres, 2% containment #HesperiaFire - 1,330, 7% #JunesFire - 1,076, 70% #JacksonFire - 876, ?% #HernandezFire - 600, 25% #MaxFire - 500, 0% #LisaFire - 350, 0% #PointFire 150, 0%" |
| "What is it like to be wiser than your Creator? Job tried that and later repented in dust and ashes. Job was a wise and blessed man of faith." |
| "What was the true history behind all of these melted ruins? Certainly not the result of erosion, but perhaps a plasma storm? The results of a sudden flip in our polar magnetic electric field perhaps." |
| "WAVELAND, Mississippi A boil advisory was issued Mon afternoon dt burst in the main water line.
NASHVILLE, Indiana A boil advisory was issued Mon after a water main break on Honeysuckle Ln.
MONTICELLO, Kentucky A boil advisory was issued Mon for the East Hwy 92 area dt a water main break.
MILAN, Ohio Seminary Rd bw Perrin Rd/Broad St in Milan is closed dt a water main break.
SASKATOON, SK - Canada North Park Wilson School is closed dt a water main break.
NEW CASTLE CO., Delaware Shipley Rd at Foulk Rd was closed Mon night dt a water main break.
MONTGOMERY CO., Maryland Dt a water main break Mon, a portion of the NW Branch Stream Valley Park near Highwood Terrace was undergoing repairs.
SIOUX FALLS, South Dakota A water main break Mon afternoon caused flooding in downtown.
BOURBON CO., Kansas There is a water main break at the Bourbon Co Transfer Station.
FORTUNA, California Water will be shut off Wed to repair a broken water main on S Fortuna Blvd." |

**Table D.9: Top-5 posts by total engagement for cross-platform coordinated accounts**

| Topic | Representative Tweets |
| --- | --- |
| rfk | "If Kennedy iselected it will cost the evil doers over a trillion dollars in the first year. If Kennedy is in the debate it will show Biden as sinile, Trump as weak and Bobby will be elected. So they won't let Bobby in but will compensate CNN and anyone that might go to jail."
"I've never been more convinced that both Trump and Biden fear Kennedy. The uniparty has no soul, and they hate democracy. We will never forget their attempts to silence us. Kennedy FTW" |
| debate | "Biden's really not up to debate is what this means. Whenever a question is asked, Biden's promoters will either tell the answer in an earplug or use a prompter screen on the podium so Biden can read the answer. This is funny."
"Trump has guts though. They tried to make it as unfriendly for him as possible, and he still is going to do it. I'm sure they were hoping he would say no way, and then Biden could brag about Trump being afraid to debate him." |
| covid-vaccine | "Remember this on election day, remember All the "suddenly died" family and friends! Biden did this, he made vaccines mandatory! Just one of many bad Biden decisions!"
"Trump was during Pandemic; FAILED to protect the American people when he knew damn well how infectious; deadly Covid Virus was, instead Trump told America "Covid was a Hoax"; failed to create a National Suppression Plan to save lives 700KAmericans DIED under Trump!" |
| bowman-latimer | "Bowman called Biden a liar. Bowman voted against the Infrastructure Act. Bowman even voted against the Debt Limit increase that could have jeopardized Social Security. Bowman did not earn another term as a Democrat."
"Bowman is his worst enemy. His and Rape denying has turned his district against him. His Votes AGAINST the Biden Administration's Progressive policies are NOT helping his district! Vote !" |
| border-security | "with your lies. It is that BLOCKED Bipartisan Border Security Bill bc told GOP to "Kill The Bill". publicly stated it. The Border Crisis is NOW all on for Blocking Border Bill to SECURE THE BORDER!"
"This bill was a sham and would not close border,but would allow more to take American jobs Donalds reiterates why GOP rejected 'bipartisan' border bill to head off potential debate talking point" |
| epstein-files | "His flights with Epstein were with family present and years before he had an island. He's also the only one saying anything about releasing the logs, Biden sure isn't, and Trump took umbrage when DeSantis started advocating for their release for some strange reason..."
" just another day in 'Murica, home of the child sex slaves. Trump made these folks feel they deserved Matt Gaetz' job, Trumps' job or Epstein's job if they just 'networked' enough. Investigate the real 'satanic' sex cult that police and politicians hide." |
| student-loan | "This administration and president Biden seem to be answerable to nobody between giving money to student loan forgiveness even after the court ruled against them not getting Kennedy secret service protectiin, the list goes on no accountability."
""Summary: We estimate that President Bidenś recently announced "New Plans" to provide relief to student borrowers will cost 84$billion, in addition to the$475 billion that we previously estimated for President Bidenś SAVE plan."" |
| hunter-biden | "Hunter Biden's laptop has nothing to do with the false Electoral College votes sent in for the 2020 election. Also, Alexander Smirnov (an FBI informant) was INDICTED for his false accusations about Barisma and the Bidens."
"Actions speak louder than how they phrased something – especially when the actions are repeated over and over again. Their lies are proven in this week's Hunter Biden trial where the laptop is used as evidence." |
| blacks-against-biden | "People aren't paying attention and don't expect him to be the President and don't know about Biden's accomplishments. That's what the polls show now but that's gonna change. Trump isn't gonna win 20% of the black vote. No were here near it"
"'Bully and terrorize and bribe': 'Deeply alarmed' conservative issues red-alert warning Former President appears in Nazi Dictator Adolf Hitler's visor cap in the photo illustration above a plea for sanity ... SICK!!!" |

**Table E.10: Topics and their representative posts on Twitter**

