# OpenReview forum: "Exposing Cross-Platform Coordinated Inauthentic Activity in the Run-Up to the 2024 U.S. Election"
_ACM.org/TheWebConf/2025/Conference — WWW 2025 Oral_

### Official Review · Reviewer_VDPV · 2024-11-25

**Novelty:** 5
**Technical Quality:** 5

**Review:**

Thank you for the opportunity to review this manuscript. This research presents a new method that uses network analysis to identify accounts acting together in coordinated ways across different social media platforms (Twitter, Telegram and Facebook). The work presents several interesting findings, though there are some areas that could be strengthened.

The paper is well written and organized, making it easy for readers to follow the research narrative and methodology.

The figures effectively complement the authors' narrative, providing visual support for their findings and arguments.

The research methodology demonstrates competent use of diverse state-of-the-art technologies in preprocessing and other part of the analysis.

The investigation of cross-platform inauthentic behavior represents a novel contribution to the field, addressing an important and timely issue.

The authors maintain transparency throughout the paper by openly discussing the limitations of their analytical approaches.

The usage of a null model would strengthen the validity of the results and tackle the issue regarding the discrepancy of dynamics of different platforms. This is pointed out by the authors themselves.

Some type of sensitivity analysis in filtering based on similarities and eigenvector values would have made the findings slightly more robust. While the authors mention that the 0.95 and 0.5% are standard in the literature, it might be a good idea to do a recheck especially as we are exploring multiple platforms this time.

**Questions:**

1) In abstract, you claim that “despite platform efforts to curb them.”  What efforts are you specifically referring to?

2) You mention “false narratives” on line 184. You might want to be careful with this term. Terms, which you already use in your paper, such as highly partisan, low-credibility, conspirational content, are more specific, hence perhaps more preferable? Narratives can be misleading, confusing, implausible, etc. but stamping any narrative directly false might be too radical in research. What do you think?

3) Is there a reason why you decided to use cosine similarity when assessing the similarities of interactions? Do you assume there are some separate isolated clusters in the vector space where cosine similarity outperforms simpler measures such as the Jaccard Index or Euclidean distance? What about taking into account the magnitude (although it is not likely that a user reposts exact same urls again)?

4) You mention on line 843 “We observe that the intersection between these two patterns is modest, with overlap rates”. I assume you are referring to accounts here, right? So the conclusion would be that distinct tactics are employed by different set of users?

4) I personally found the final part of the paper on “engagement” vague. I am still not sure what “engagement is being generated” here? Is it the number of reached users, likes, reposts? Something else? I think this should be further clarified, because it made interpretting the ECDF-figures almost impossible.

Some minor stuff;

LINE 124: coIA -> CoIA for consistency?
LINE 366: Many “Appendix” refs in the main text did not contain the corresponding section number.
LINE 589: Missing Table number.
LINE 822: Table -> Figure in the caption?

I enjoyed reading the paper - I hope you find my comments constructive!

**Reviewer Confidence:**

3: The reviewer is confident but not certain that the evaluation is correct

**Scope:**

4: The work is relevant to the Web and to the track, and is of broad interest to the community

---

### Official Review · Reviewer_Ue3Z · 2024-11-29

**Novelty:** 6
**Technical Quality:** 6

**Review:**

Pros:
- The paper is well written and structured, and easy to follow.
- The research is well motivated and references to the literature are relevant and extensive enough.
- The research questions are well formulated and it is clear how each part of the analysis answers each question.
- The paper is technically and theoretically sound.

Cons:
- Perhaps the discussion could be slightly extended. For instance, the implications of the contributions can be emphasised more. What can this methodology achieve that couldn't be done otherwise? What insights did the analysis provide that are novel compared to what has already been recorded in the literature?
- The paper is about coordinated inauthentic activity. While the focus on the coordination is very clear, the focus on inauthenticity isn't as clear. From my reading of the paper, there isn't much investigation regarding the authenticity of the accounts (i.e. are they really who they claim to be?). My understanding is that coordination is the single criterion used to label users as inauthentic. I'm not entirely convinced this is sufficient to claim inauthenticity. I would recommend minimising claims about the inauthenticity of these accounts in the manuscript (e.g. by clearly stating that the interpretation of inauthenticity is speculative) and focusing more on the idea of coordination, which the paper provides plenty of evidence of.
- The description of the method used for tuning the threshold for the node and edge pruning ("Density-based network dismantling") is unclear to me (see questions below). The description of this method should be made clearer/extended.
- Under Section 5.1: You should report what the proportion of users on each platform your method has classified as organic vs coordinated. This information is important for interpreting other results (e.g. the prevalence of AIGC on each platform by organic vs coordinated users).

Formatting/writing points:
- p. 2 l.202: What does "IOs" mean here? This needs to be clarified.
- p.3 Table 1: This table can be improved to avoid the repetition of headers. Moreover, it would be interesting to include the ratio of posts to accounts, of URLs to posts and of domains to URLs. Perhaps this can be added in the Appendix.
- p.5: When you mention "Table D.2", "Table D.6", etc., I struggled to understand which tables you were referring to. I would suggest stating explicitly these tables are in the Appendix.
- p.6 l.589: The reference to the table is missing.

**Questions:**

- For the co-URL similarity network, my understanding is that you are looking for only at exact URL matches. However different URLs  might point to very similar material. Have you considered looking at the similarity in the content from the URLs? If not, how do you think this would add to/change your analysis?
- p.3 regarding the user similarity vectors: How large are these user vectors? My understanding that this will depend on the number of unique URLs in the dataset. If each dimension corresponds to a single URL the vectors must be very sparse. Did you use any dimensionality reduction? Moreover, if the vector sizes are indeed different for the different datasets, how does this impact your results? I'm particularly curious about the effect this might have on the "fully-connectedness" measure you use.
- p.3 "In particular, we use network density as a variable to modulate the combination of thresholds for both node and edge filtering". I'm not sure I fully understand how you do this. You say you look for a "transitional phase" in the network density as you perform a grid search on the thresholds for the node and edge pruning. What does this mean exactly?
- p. 5 Figure 1: If I understand correctly this visualisation is obtained from the co-URL network. Have you tried plotting a similar visualisation using the content amplification network? Is it similar? Are there noteworthy differences? Would it be useful to merge the two  (e.g. by average or adding the two similarity measures)?
- Also about Figure 1: What do the percentages in the bottom right represent? My understanding is that this is the proportion of the inter-platform network corresponding to each platform but it isn't clear
- p.6: Some analysis of user descriptions is provided here. This would have been a good place to investigate the question of "authenticity": who do these account claim to be and, in your opinion, does the rest of the results you have gathered constitute sufficient evidence that this is not who they truly are?
- p.7 l.789: "No conspiracy theories were detected among the coordinated users" --> This is interesting. To what extent did you try to verify this claim? I'm wondering for instance whether conspiracy theories about covid or covid vaccines might not be included as a sub-topic under the "covid-vaccine" topic on Twitter. It might also be useful if you provide a clearer definition of what you consider a conspiracy theory as opposed to misinformation/disinformation.
- p.8 Figure 5: This figure shows the prevalence of AI generated content on different platforms and amongst organic vs coordinated users. How are these numbers calculated? Is it the percentage of posts? Or the percentage of users who post AIGC? Or is it the prevalence of all the content in that dataset? This is important and should be compared to the prevalence of organic vs coordinated users on each platform.

**Reviewer Confidence:**

3: The reviewer is confident but not certain that the evaluation is correct

**Scope:**

4: The work is relevant to the Web and to the track, and is of broad interest to the community

---

### Official Review · Reviewer_PAbk · 2024-12-02

**Novelty:** 4
**Technical Quality:** 4

**Review:**

By constructing similarity networks through shared URLs and content similarity on the 2024 U.S. election data from Telegram, X, and Facebook between May and June 2024, the authors detected and analyzed coordinated communities within and across platforms. They found that URLs and content shared by these communities were typically highly partisan, low-credibility, and conspiratorial. These findings suggest the need for regulatory measures to address cross-platform coordination.

This study contributes to research on coordinated activities by exploring cross-platform coordination.
However, the manuscript would benefit from clearer explanations of key terms and stronger connections between the research questions, methods, and inferences, as I elaborate below.

General Suggestions:
Enhance the explanations of key terms, especially "coordinated inauthentic activity," to ensure clarity for the readers.

Strengthen the connection between research questions, methods, and findings, particularly in the context of AI-generated content.

Align the domain analysis methodology with standard practices by excluding irrelevant domains (e.g., apps, search engines, and social media platforms) during preprocessing.

Provide more detail on the suggested regulatory measures beyond individual platforms. Specifically, consider incorporating the observed differences in shared content and URL features across platforms to inform these recommendations.

**Questions:**

Coordinated Inauthentic Activity:
The authors use the term "coordinated inauthentic activity," but it is unclear how they identified these activities as "inauthentic." Providing explicit criteria or examples of inauthentic coordination would strengthen the manuscript's claims.

AI-Generated Content Across Platforms:
The authors detected AI-generated content and observed interesting differences across platforms. However, they do not explain the motivation for detecting AI-generated content, how it relates to their three research questions, or how to interpret the observed differences. Clarifying these points would improve the coherence and impact of the findings.

Data Collection Period:
While the authors specify the data collection period, they do not explain how far back the data traces. For instance, the prominence of vaccine skepticism and holistic health on Telegram suggests that some data may predate the specified collection window. It would be helpful if the authors could clarify the earliest available timestamps for their dataset.

Top Shared Domains in Table 4:
The domain "truthsocial.com" is categorized as "not partisan," which seems imprecise given the ideological motivations behind its creation. MBFC does not rate its ideological leaning because it is a social media platform. Misinformation studies typically exclude search engines and social media platforms from domain analyses. While it is acceptable to include "truthsocial.com" in the analysis, the authors should reconsider its description.

Engagement with Low-Credibility Information:
Studies have shown that conservative individuals tend to generate more engagement by sharing low-credibility information. The authors may find the following reference helpful to contextualize this observation:
Biswas, A., Lin, Y. R., Tai, Y. C., & Desmarais, B. A. (2024). Political Elites in the Attention Economy: Visibility Over Civility and Credibility? arXiv preprint arXiv:2407.16014.

Cross-Platform Coordination and Russian State-Affiliated Websites:
The claim of "potential Russian information operations" is not justified without providing the number of shares for the seven Russian state-affiliated websites. More detailed data is required to support such an assertion.

Top Shared Domains on Twitter:
The domains shared on Twitter differ from those shared on Telegram in terms of sharing volume and their low-credibility or conspiratorial features. For example, the frequent sharing of "foxnews.com" could be driven by the established conservative media ecosystem rather than coordinated promotion. The following reference may help contextualize this observation:
Tai, Y. C., Buma, R., & Desmarais, B. A. (2024). Official Yet Questionable: Examining Misinformation in U.S. State Legislators’ Tweets. Journal of Information Technology & Politics, 21(4), 597-609.

Additionally, "newsbreakapp" and "go.shr.lc" should be excluded from the domain analysis since they are apps and URL shorteners. Generally, search engines, social media platforms, and apps should be removed during preprocessing to ensure the focus remains on relevant domains. This practice is well-documented:
Lasser, J., Aroyehun, S. T., Simchon, A., Carrella, F., Garcia, D., & Lewandowsky, S. (2022). Social Media Sharing of Low-Quality News Sources by Political Elites. PNAS Nexus, 1(4), pgac186.

Cross-platform regulatory:
While the authors propose regulatory measures beyond individual platforms, they do not elaborate on the specific types of regulations needed. Considering the observed differences in shared content and URL characteristics across platforms, these variations should be considered when any recommendations for cross-platform regulations are made.

**Reviewer Confidence:**

3: The reviewer is confident but not certain that the evaluation is correct

**Scope:**

4: The work is relevant to the Web and to the track, and is of broad interest to the community

---

### Official Review · Reviewer_FeGS · 2024-12-03

**Novelty:** 5
**Technical Quality:** 6

**Review:**

This paper analyses both intra- and cross-platform coordinated inauthentic activity on X, Facebook, and Telegram, focusing in particular on the 2024 U.S. Election. By leveraging different detection models and a large-scale dataset, the research focuses on uncovering evidence of coordinated efforts, highlighting the need for regulatory measures to address the cross-platform coordinated influence campaigns.

**Pros**
- The paper is well-written and easy for readers to follow.
- The paper provides a thorough analysis from multiple perspectives, while at the same time the authors justify the different choices they made.
- The authors will make the code and data publicly available upon acceptance.

&nbsp;

**Comments**
- It would be nice if the authors summarised in the Related Work section the advantages/differences of this study compared to the state-of-the-art.

&nbsp;

**Minor comments**
- Please use either CoIAs or coIAs throughout the paper.
- I believe it would be better to move the two sentences in lines 331–335 to a new paragraph and merge them with the next paragraph (lines 336–343).
- Figure 1: “Cross-platform coordination network showing user coordination across four social media networks.” -> “Cross-platform coordination network showing user coordination across three social media networks.”
- (line 551) “As shown in Table D.2, the most frequently shared domains …” -> “As shown in Table 2, the most frequently shared domains …”
- When there is a reference to a table in the Annex, please refer to it specifically in the text. E.g. Instead of "see Table D.2" say "see Table D.2 in the Annex" or something along these lines.
- (line 589) “The top-engagement messages (see Table D.8) and bios (see Table ??)”: The bios Table is missing from the ANNEX.
- Table 6: This appears to be a figure and should be labeled as such.
- Under Section 5.3, please add a reference to RQ3.

**Questions:**

1. Do the authors intend to make the code and data publicly available?

**Reviewer Confidence:**

3: The reviewer is confident but not certain that the evaluation is correct

**Scope:**

4: The work is relevant to the Web and to the track, and is of broad interest to the community